# STAS: Spatio-Temporal Adaptive Computation Time for Spiking Transformers

## Abstract

Spiking neural networks (SNNs), while energy-efficient, suffer from high latency and computational overhead, and existing dynamic computation methods to address this remain fragmented. While the principles of adaptive computation time (ACT) offer a robust foundation for a unified approach, its application to SNN-based vision Transformers (ViTs) is hindered by two core issues: the violation of its temporal similarity prerequisite and a static architecture fundamentally unsuited for its principles. To address these challenges, we propose STAS (**S**patio-**T**emporal **A**daptive computation time for **S**piking transformers), a framework that co-designs the static architecture and dynamic computation policy. STAS introduces an integrated spike patch splitting (I-SPS) module to establish temporal stability by creating a unified input representation, thereby solving the architectural problem of temporal dissimilarity. This stability, in turn, allows our adaptive spiking self-attention (A-SSA) module to perform two-dimensional token pruning across both spatial and temporal axes. Implemented on spiking Transformer architectures and validated on CIFAR-10, CIFAR-100, and ImageNet, STAS reduces energy consumption by up to 45.9%, 43.8%, and 30.1%, respectively, while simultaneously improving accuracy over SOTA models.

## 1 Introduction

Spiking neural networks (SNNs) are energy-efficient but suffer from high latency and computational overhead due to their multi-timestep operational nature. State-of-the-art (SOTA) studies to improve SNNs have followed two main paths: **(S)** static architectural enhancements (e.g., Spikformer (Zhou et al., 2022), Spikingformer (Zhou et al., 2023)) and **(D)** dynamic computation methods (e.g., OST (Song et al., 2024), STATA (Zhuge et al., 2024)), with their performances shown in Fig. 1(a). Dynamic methods are motivated by the observation that accuracy often saturates long before the final block or timestep, presenting a clear opportunity for input-dependent halting (Fig. 1(b)).

The exploration of dynamic computation has fragmented into distinct approaches. One line of research has refined **(D1)** architecture-agnostic spatial halting (e.g., SACT (Figurnov et al., 2017)). In parallel, SNN-specific works have focused on **(D2)** temporal adaptivity (e.g., DT-SNN (Li et al., 2023)). A third approach is **(D3)** architecture-aware halting (e.g., A-ViT (Yin et al., 2022)), which leverages a model's unique components, such as Transformer tokens. These strategies operate independently along a single dimension, not only due to a lack of research into their synergy, but because of a fundamental conflict we identify (in Sec. 3.1): the direct application of a method from one dimension can degrade performance in another. This issue becomes particularly evident when powerful halting principles are applied to the unique temporal dynamics of SNNs.

This architectural conflict is aptly illustrated by the principles of adaptive computation time (ACT). While ACT offers a potential foundation for a unified framework, its direct application to SNN-based vision Transformers (ViTs) reveals a deeper, architectural obstacle. ACT's efficacy is critically constrained by the static architecture; it relies on high input similarity for stable refinement. While SNN-ViTs possess spatial similarity, their design leads to critically low temporal similarity due to varying spike inputs at each timestep. This architectural flaw makes true spatio-temporal halting impossible with a purely dynamic approach and reveals a critical interdependence: an effective dynamic framework requires a new static architecture, necessitating an integrated **(S with D1–D3)** paradigm.

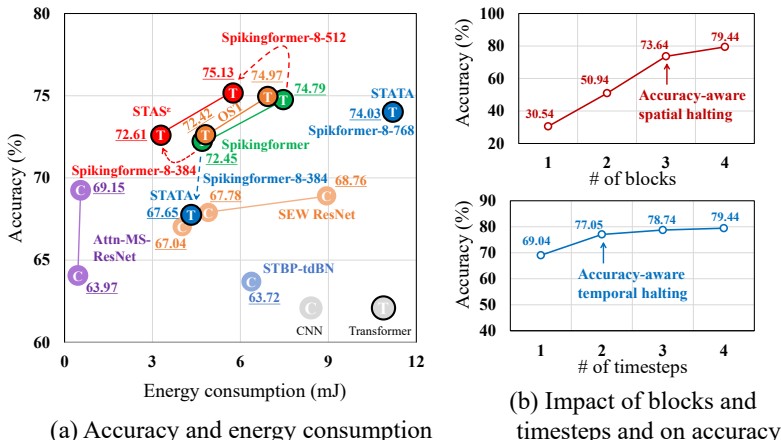

(a) Accuracy and energy consumption

(b) Impact of blocks and timesteps and on accuracy

Figure 1: Accuracy of adaptive computation methods for Spikingformer on ImageNet. (a) Accuracy-energy trade-off for various models (see Table 2). (b) Accuracy saturation motivating halting, shown spatially (top) and temporally (bottom).

In this paper, we propose STAS (**S**patio-**T**emporal **A**daptive computation time for **S**piking Transformers), a novel framework that resolves this interdependence by co-designing the static architecture and the dynamic computation method. STAS first addresses the architectural bottleneck with an integrated spike patch splitting (I-SPS) module, providing the static solution **S** by creating a temporally unified representation. This engineered stability, in turn, unlocks true two-dimensional adaptivity, enabling our adaptive spiking self-attention (A-SSA) module to act as the unified framework for **D1−D3** by performing concurrent token halting across both spatial and temporal axes.

We implemented STAS on strong, directly trained spiking Transformers, including Spikformer and Spikingformer, and validated its performance on the CIFAR-10, CIFAR-100, and ImageNet classification datasets. When applied to these architectures, STAS reduces energy consumption by up to 45.9%, 43.8%, and 30.1% on the three datasets, respectively, while simultaneously improving top-1 accuracy.

Our contribution can be summarized as follows:

- We diagnose the fundamental barrier to a unified adaptive framework in SNN-based ViTs through a spatio-temporal similarity analysis, revealing that their architectural design inherently obstructs temporal halting.

- We propose I-SPS that re-engineers the SNN input stage to establish the temporal similarity required for effective temporal adaptation.

- Building upon the stability provided by I-SPS, we introduce A-SSA, a unified mechanism that performs concurrent spatial and temporal token halting.

- We demonstrate the effectiveness of STAS through extensive experiments on CIFAR-10, CIFAR-100, and ImageNet, achieving up to 45.9%, 43.8%, and 30.1%, respectively, for SOTA architectures while improving accuracy.

## 2 RELATED WORK

Methods like DT-SNN dynamically adjust the timesteps of an SNN during inference based on accuracy needs, using entropy and confidence metrics to halt computation early for simpler inputs. SEENN (Li et al., 2023; 2024) employs reinforcement learning to optimize timesteps for each image, allowing for fine-grained per-instance optimization, while TET (Deng et al., 2022) introduces a loss function to address gradient loss in spiking neurons, achieving higher accuracy with fewer timesteps. However, the decision-making overhead of these temporal methods can outweigh the benefits in low-timestep regimes, making them less suitable for deeper, more efficient models. In a different approach, MST (Wang et al., 2023) proposes an ANN-to-SNN conversion method for SNN-based ViTs, using token masking within model blocks to reduce energy consumption. Despite

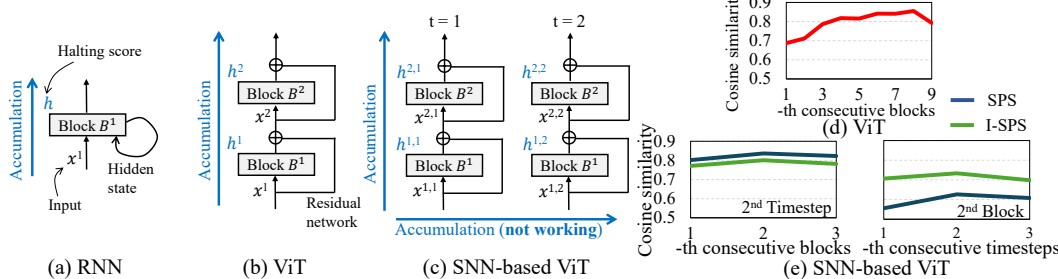

Figure 2: Model architecture and halting-score accumulation paths when Adaptive Computation Time (ACT) is applied: (a) RNN, (b) ViT, and (c) SNN-based ViT. Cosine similarity of tokens between consecutive blocks for (d) ViT and (e) SNN-based ViT (Spikingformer) on CIFAR-100.

its effectiveness, this reliance on ANN-to-SNN conversion means MST still requires hundreds of timesteps for inference.

The principles of ACT (Graves, 2016) were first introduced to dynamically allocate inference steps for RNN models based on input difficulty. This concept was extended by SACT (Figurnov et al., 2017) for ResNet architectures and A-ViT (Yin et al., 2022), which dynamically adjusts computation in Transformers by halting individual tokens at different layers. However, these studies are based on ANNs, and their formulations are fundamentally incompatible with the discrete, multi-timestep nature of SNNs, as they typically perform a single inference pass. While LFACT (Zhang et al., 2021) expands ACT for repeated inferences across sequences, it remains limited to RNNs. In contrast, STAS is explicitly designed to address the unique two-dimensional challenge of SNN-based ViTs, simultaneously considering adaptivity across both spatial blocks and discrete timesteps.

## 3 METHOD

### 3.1 I-SPS: INTEGRATED SPIKE PATCH SPLITTING

ACT enables neural networks to dynamically adjust their computational depth per input, learning to halt processing to improve efficiency. The mechanism is predicated on the principle of halting computation once the network's internal representations stabilize. This concept was originally proposed for RNNs, where an encoder block $\mathcal{B}^1$ iteratively refines its state from the same input $x^1$, and a sigmoidal halting unit determines when to cease processing (Fig. 2(a)). This architectural paradigm extends naturally to ViTs, which can be viewed as an "unrolled iterative estimation" process. Their structure, featuring multiple identical encoder blocks (property (i)) with residual connections (Fig. 2(b)), ensures high input similarity between consecutive blocks (property (ii), Fig. 2(d)). This representational stability is a prerequisite for ACT, enabling effective spatial halting in ViTs by allowing each block $\mathcal{B}^i$ to accumulate a corresponding halting score $h^i$ (Yin et al., 2022).

However, applying ACT to SNN-based ViTs introduces a dual-dimensional challenge, as the conditions for effective halting must be met across both spatial (inter-block) and temporal (inter-timestep) axes (Fig. 2(c)). **Spatially**, SNN-based ViTs are analogous to their standard counterparts; they satisfy property (i) via residual connections and, consequently, maintain high block-to-block similarity (property (ii)), making them suitable for spatial ACT (left subfigure of Fig. 2(e)). **Temporally**, the challenge is more profound. While property (i) is satisfied because membrane potentials are shared across timesteps within the same block, SNNs inherently violate property (ii). Each timestep receives a different input spike vector, leading to low cosine similarity between consecutive temporal inputs, as shown by the blue curve in the right subfigure of Fig. 2(e).

To address the low temporal similarity in SNN-based ViTs that impedes ACT, we introduce the I-SPS module. Unlike vanilla SPS, I-SPS integrates multi-timestep spike signals into a single, unified representation at the initial stage, which is then reused for all subsequent computations (Fig. 3(b)). This positions our method as a type of 'one-step' approach[1], an emerging concept in SOTA SNN

---

[1]This is termed a 'one-step' approach because the computationally expensive CNN operation is reduced to a single pass, while the low-latency LIF neuron operations still iterate for $T$ timesteps.

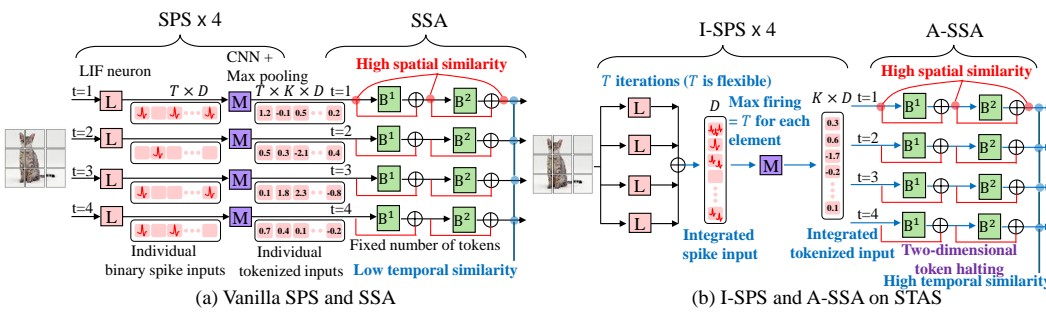

(a) Vanilla SPS and SSA

(b) I-SPS and A-SSA on STAS

Figure 3: Architectural comparison of (a) a conventional SNN-based ViT using vanilla SPS and SSA, and (b) our STAS framework featuring I-SPS and A-SSA. STAS utilizes I-SPS to create a single, unified tokenized input from multiple timesteps, which establishes the high temporal similarity necessary for the two-dimensional token halting performed by A-SSA.

Table 1: Effectiveness of I-SPS for A-SSA on Spikformer-4-384 and Spikingformer-4-384 with CIFAR-100.

| Architecture | I-SPS | A-SSA | Avg. tokens | Acc (%) |
|---|---|---|---|---|
| Spikformer | ✗ | ✗ | ×1 | 77.3 |
| | ✗ | ✓ | ×0.63 | 77.3 (−) |
| | ✓ | ✓ | ×**0.46** | **78.1** (↑) |
| Spikingformer | ✗ | ✗ | ×1 | 79.4 |
| | ✗ | ✓ | ×0.95 | 77.4 (↓) |
| | ✓ | ✓ | ×**0.70** | **79.9** (↑) |

studies where expensive operations are reduced to a single pass in distinct ways for varied goals, such as latency reduction (e.g., OST) or simplified adversarial attacks (e.g., RGA (Bu et al., 2023)). The viability of such methods, which sacrifice precise temporal information, is rooted in mitigating challenges in direct SNN training; a shortened temporal backpropagation path reduces the impact of both vanishing gradients and error accumulation from surrogate functions. This improved gradient flow appears to offset the information loss from temporal compression. STAS operationalizes this principle via the I-SPS module, creating the high temporal similarity (Fig. 2(e)) that is the prerequisite for our A-SSA module to perform dynamic, two-dimensional token halting.

**Empirical validation.** Table 1 validates the synergistic relationship between our static architectural module (I-SPS) and dynamic halting mechanism (A-SSA), which is detailed in Sec. 3.2. Applying A-SSA alone is ineffective, yielding only a limited token reduction on both Spikeformer and Spikingformer (×0.63 and ×0.95, respectively). However, when combined with I-SPS—which establishes the necessary temporal similarity—the synergy drastically reduces token usage to ×0.46 on Spikformer and ×0.70 on Spikingformer, while maintaining or even slightly improving accuracy. These results empirically demonstrate that I-SPS is a critical prerequisite for A-SSA to perform efficient and accuracy-aware spatio-temporal halting.

## 3.2 A-SSA: ADAPTIVE SPIKING SELF-ATTENTION

We formulate the SNN-based ViT as follows (Zhou et al., 2023):

$$f_T(x) = FC(\frac{1}{T} \sum_{t=1}^{T} \mathcal{B}^L \circ \mathcal{B}^{L-1} \circ \cdots \circ \mathcal{B}^1 \circ \mathcal{S}(x)), \tag{1}$$

where $x \in \mathbb{R}^{T \times C \times H \times W}$ is the input of which $T$, $C$, $H$, and $W$ denote the timesteps, channels, height, and width.

The function $\mathcal{S}(\cdot)$ represents the spike patch splitting (SPS) module, which divides the input image into multiple tokens. The function $\mathcal{B}(\cdot)$ denotes a single encoder block, consisting of spike self-attention (SSA) and a multi-layer perceptron (MLP), with a total of $L$ blocks in the model. The

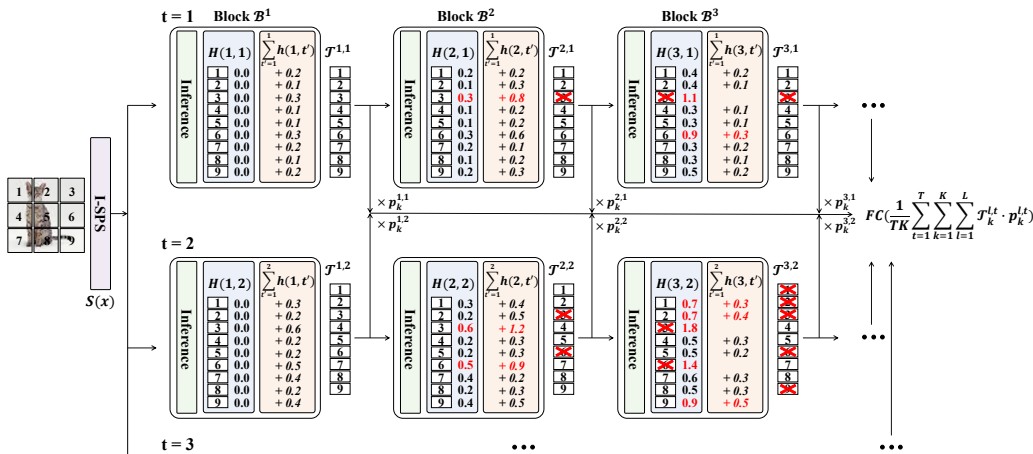

Figure 4: Token-level halting example of STAS: At the first timestep $t = 1$, the input $x$ passes through the I-SPS, generating a token set $\mathcal{T}^{l,t}$. In the first block $\mathcal{B}^1$, for nine tokens, the halting scores $h_k^{1,1}$ are added through inference. In subsequent blocks, tokens with accumulated halting scores $H(l,t)$ of one or greater are masked. From the second timestep onwards, the same operations are repeated on the same input $x$. The halting score accumulation follows Eq. equation 4. The vector values of masked tokens are set to zero, and no further halting score is accumulated for the tokens.

function $FC(\cdot)$ represents a fully-connected layer. Finally, the tokens passing through all blocks are averaged and input to $FC(\cdot)$.

After passing through $\mathcal{S}(x)$ at a timestep $t$, the input image $x$ is divided into a set of tokens denoted by $\mathcal{T}^{0,t}$. Let $\mathcal{T}^{l,t}$ represent the set of tokens in the $l$-th (for $l > 0$) block at the $t$-th timestep, which is expressed as follows:

$$\mathcal{T}^{l,t} = \mathcal{B}^l(\mathcal{T}^{l-1,t}). \tag{2}$$

The halting score $h^{l,t}$ of the tokens at the $t$-th timestep in the $l$-th block can be defined as follows:

$$h_k^{l,t} = \sigma(\alpha \times \mathcal{T}_{k,1}^{l,t} + \beta), \tag{3}$$

where $\sigma(\cdot)$ denotes the logistic sigmoid function, and $\alpha$ and $\beta$ are scaling factors.

Let $\mathcal{T}_k^{l,t}$ represent the embedding vector of the $k$-th token, and $\mathcal{T}_{k,1}^{l,t}$ denote the first element of this vector. The sigmoid function ensures that $0 \leq h_k^{l,t} \leq 1$. STAS calculates $h_k^{l,t}$ using the first element of the embedding vector of the token, and the first node of MLP in each block learns the halting score.

STAS accumulates halting scores across blocks within a single timestep and continues to accumulate scores from previous timesteps and blocks over multiple timesteps, as a two-dimensional halting policy. STAS defines the halting module $H_k(L', T')$ at the $T'$-th timestep and the $L'$-th block as follows:

$$H_k(L', T') = \sum_{l=1}^{L'-1} \sum_{t=1}^{T'} h_k^{l,t}. \tag{4}$$

STAS masks tokens with $H_k(L', T') \geq 1 - \epsilon$ in each block. If the $k$-th token is halted at the $L'$-th block and $T'$-th timestep, it remains zeroed out from the $L' + 1$ block onward in the $T'$-th timestep. Fig. 4 illustrates a token-level merging and masking example of AT-SNN.

Based on the defined halting score, we propose a new loss function that allows STAS to determine the required number of tokens according to the input image during training. We define $\mathcal{N}_k^t$ as the index of the block where the $k$-th token halts at the $t$-th timestep, which is obtained by

$$\mathcal{N}_k^t = \underset{l \leq L}{\arg\min} \, H_k(l, t) \geq 1 - \epsilon, \tag{5}$$

where $\epsilon$ is a constant value that determines the threshold for the halting score.

Additionally, we define an auxiliary variable, remainder, to track the remaining amount of halting score for each token until it halts at each timestep and layer as follows:

$$r_k^{l,t} = 1 - H_k(l,t). \tag{6}$$

Then, we define the halting probability of each token at each timestep and block as follows:

$$p_k^{l,t} = \begin{cases} h_k^{l,t} \text{ if } t = \{1,...,T\} \text{ and } l < \mathcal{N}^t \\ r_k^{l,t} \text{ if } t = \{1,...,T\} \text{ and } l = \mathcal{N}^t \\ 0 \text{ otherwise} \end{cases} \tag{7}$$

According to the definitions of $h_k^{l,t}$ and $r_k^{l,t}$, $0 \leq p_k^{l,t} \leq 1$ holds.

Based on the previously defined halting module and probability, we propose the following loss functions for training STAS. First, we apply a mean-field formulation (halting-probability weighted average of previous states) to the output at each block and timestep, accumulating the results. Therefore, the classification loss function $\mathcal{L}_{task}$ is defined as follows:

$$\mathcal{L}_{task} = \mathcal{C}(FC(\frac{1}{TK}\sum_{t=1}^{T}\sum_{k=1}^{K}\sum_{l=1}^{L}\mathcal{T}_k^{l,t} \cdot p_k^{l,t})), \tag{8}$$

where $\mathcal{C}$ denotes the cross-entropy loss.

Next, we propose a loss function to encourage each token to halt at earlier timesteps and blocks, using fewer computations. We defined the ponder loss $\mathcal{L}_{ponder}$ as follows:

$$\mathcal{L}_{ponder} = \frac{1}{TK}\sum_{t=1}^{T}\sum_{k=1}^{K}(\mathcal{N}_k^t + r_k^{\mathcal{N}_k^t,t}). \tag{9}$$

$\mathcal{L}_{ponder}$ consists of the average number of blocks over which each token accumulates its halting score and the average remainder at each timestep.

$$\mathcal{L}_{overall} = \mathcal{L}_{task} + \delta_p\mathcal{L}_{ponder}, \tag{10}$$

where $\delta_p$ is a parameter that weights $\mathcal{L}_{ponder}$. STAS is trained to minimize $\mathcal{L}_{overall}$.

### 3.3 FLEXIBLE HALTING THRESHOLD

STAS adaptively determines the number of tokens to halt for each input image during training. However, during inference, there remains a trade-off between the number of tokens to halt and accuracy. To address this, we introduce STAS$^\epsilon$, a method that provides control-knob between the number of tokens to halt and accuracy by adjusting the halting threshold parameter $\epsilon$ during inference. By increasing the value of $\epsilon$, STAS$^\epsilon$ halts more tokens at earlier blocks or timesteps, leading to reduced energy consumption and accuracy.

## 4 EXPERIMENTS

We first analyze the qualitative and quantitative results to assess how efficiently STAS reduces tokens for the input images. Then, we conduct a comparative analysis to evaluate how effectively STAS reduces tokens in terms of accuracy, comparing it with existing methods, and analyze how the reduced tokens by STAS impact energy consumption. Finally, we discuss the properties required for STAS's two-dimensional ACT to efficiently process tokens through an ablation study.

**Implementation details.** We implement the simulation on Pytorch and SpikingJelly (Fang et al., 2023). All experiments in this section are conducted on Spikformer (Zhou et al., 2022) and Spikingformer (Zhou et al., 2023) on RTX NVIDIA A6000 GPUs. Note that STAS is applicable to other SNN-based vision Transformers with direct training. We first train each model by replacing its original SPS module with the proposed I-SPS, and use the resulting model as a pre-trained model for

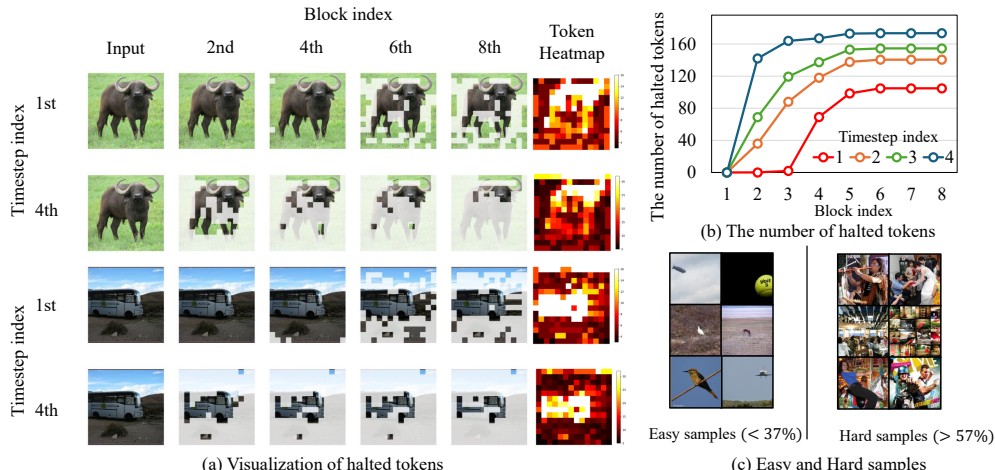

Figure 5: (a) Example of halted tokens across different timesteps and blocks on STAS$^\epsilon$ (based on Spikingformer-8-384) with ImageNet. Tokens that are halted with a shaded (non-white) overlay. (b) The number of halted tokens across different blocks and timesteps, and (c) visual comparison of hard and easy samples in classification on STAS$^\epsilon$ (based on Spikingformer-8-384) with ImageNet.

applying the proposed two-dimensional ACT. Subsequently, we retrain the models using the loss function defined in Eq. equation 10. We use automatic-mixed precision (AMP) (Micikevicius et al., 2017) for training acceleration and surrogate module learning (SML) (Deng et al., 2023) method to mitigate the gradient errors of SNNs. For a fair comparison, we trained several existing methods (e.g., Spikformer, Spikingformer, and STATA[2]) on our computing environment, and these models are marked with an asterisk (*) in Tables 2 and 3. We evaluate our method for the classification task on CIFAR-10, CIFAR-100 (Krizhevsky et al., 2009), and ImageNet (Deng et al., 2009).

## 4.1 ANALYSIS

**Qualitative results.** For visualization of STAS$^\epsilon$, we use Spikingformer-8-384 with eight blocks per timestep, trained on ImageNet. Each input image contains 196 tokens ($14 \times 14$). Fig. 5(a) visualizes how tokens are halted over timesteps and blocks. Since STAS accumulates halting scores in two dimensions (blocks and timesteps), more tokens are halted as the block and timestep indices increase. With four timesteps and eight blocks, the maximum processed count for each token is 32, where brighter regions indicate more processing, and darker regions indicate less (i.e., halted earlier). Tokens from the less informative background are halted first, with an increasing number of tokens being halted over time.

**Quantitative results and classification difficulty.** Fig. 5(b) shows the number of tokens halted per block and timestep. As visualized in Fig. 5(a), more tokens are halted as the block and timestep indices increase. Due to the two-dimensional halting policy of STAS$^\epsilon$, more tokens halt as the number of timesteps increases. Figure 5(c) visualizes samples correctly classified by STAS$^\epsilon$, comparing those that use more tokens versus those that use fewer tokens. On average, easy samples utilize 37% or fewer of all tokens per block, while hard samples use 57% or more of all tokens per block. We observe that STAS$^\epsilon$ uses fewer tokens when the object in the image is clearly separated from the background and other objects.

## 4.2 COMPARISON TO PRIOR ART

We evaluate STAS against SNN methods based on both CNNs (e.g., VGG, ResNet) and Transformers (e.g., Spikformer, Spikingformer). To benchmark against other dynamic computation techniques

---

[2]As the official implementation is not publicly available, we re-implemented the method based on the descriptions in the original paper and made our best effort to reproduce it faithfully.

Table 2: Main experiment results on ImageNet.

| Method | Architecture | Param (M) | Timestep | Energy (mJ) | Acc |
|---|---|---|---|---|---|
| Hybrid training (Rathi et al., 2020) | ResNet-34 | 21.79 | 250 | - | 61.48 |
| STBP-tdBN (Zheng et al., 2021) | ResNet-34 | 21.79 | 6 | 6.39 | 63.72 |
| TET (Deng et al., 2022) | Spiking-ResNet-34 | 21.79 | 6 | - | 64.79 |
| | SEW ResNet-34 | 21.79 | 4 | - | 68.00 |
| Spiking ResNet (Hu et al., 2021a) | ResNet-34 | 21.79 | 350 | 59.30 | 71.61 |
| | ResNet-50 | 25.56 | 350 | 70.93 | 72.75 |
| | SEW ResNet-34 | 21.79 | 4 | 4.04 | 67.04 |
| SEW ResNet (Fang et al., 2021) | SEW ResNet-50 | 25.56 | 4 | 4.89 | 67.78 |
| | SEW ResNet-101 | 44.55 | 4 | 8.91 | 68.76 |
| | SEW ResNet-152 | 60.19 | 4 | 12.89 | 69.26 |
| MS-ResNet (Hu et al., 2021b) | ResNet-104 | 44.55+ | 5 | - | 74.21 |
| Att MS ResNet (Yao et al., 2023) | Att-MS-ResNet-18 | 11.87 | 1 | 0.48 | 63.97 |
| | Att-MS-ResNet-34 | 22.12 | 1 | 0.57 | 69.15 |
| ANN | Transformer-8-512 | 29.68 | - | 38.34 | 80.80 |
| Spikformer (Zhou et al., 2022) | Spikformer-8-768 | 66.34 | 4 | 21.48 | 74.81 |
| OST (Song et al., 2024) | OST-8-384 | 19.36 | 1 | 4.63 | 72.42 |
| | OST-8-512 | 33.87 | 1 | 6.92 | 74.97 |
| Spikingformer (Zhou et al., 2023) | Spikingformer-8-384 | 16.81 | 4 | **4.69** | **72.45** |
| | Spikingformer-8-512 | 29.68 | 4 | **7.46** | **74.79** |
| STATA (Zhuge et al., 2024) | Spikingformer-8-384 | 16.82 | 4 | 4.33* | 67.65* |
| | Spikformer-8-768 | - | 4 | 11.16 | 74.03 |
| STAS | Spikingformer-8-384 | 16.81 | 4 | **3.81 (-18.8%)** | **73.45 (↑)** |
| | Spikingformer-8-512 | 29.68 | 4 | **7.16 (-4.02%)** | **75.96 (↑)** |
| STAS$^\epsilon$ | Spikingformer-8-384 | 16.83 | 4 | **3.28 (-30.1%)** | **72.61 (↑)** |
| | Spikingformer-8-512 | 29.68 | 4 | **5.73 (-23.19%)** | **75.13 (↑)** |

Table 3: Experiment results on CIFAR-10/CIFAR-100.

| Method | Architecture | Param (M) | Timestep | Energy (mJ) | Acc |
|---|---|---|---|---|---|
| STBP-tdBN (Zheng et al., 2021) | ResNet-19 | 12.63 | 4 | - | 92.9/70.9 |
| AutoSNN (Na et al., 2022) | AutoSNN (C=128) | 21 | 8 | - | 93.2/69.2 |
| SpikeDHS$^D$ (Che et al., 2022) | SpikeDHS-CLA (n3s1) | 14 | 6 | - | 95.4/76.3 |
| Hybrid training (Rathi et al., 2020) | VGG-11 | 9.27 | 125 | - | 92.2/67.9 |
| Diet-SNN (Rathi & Roy, 2020) | ResNet-20 | 0.27 | 10/5 | - | 92.5/64.1 |
| TET (Deng et al., 2022) | ResNet-19 | 12.63 | 4 | - | 94.4/74.5 |
| ANN-to-SNN (Deng & Gu, 2021) | ResNet-20 | 10.91 | 32 | - | 93.3/68.4 |
| ANN | Transformer-4-384 | 9.32 | - | 4.25 | 96.7/81.0 |
| Spikformer (Zhou et al., 2022) | Spikformer-4-384 | 9.32 | 4 | **0.74*/0.89*** | **94.8*/77.3*** |
| STATA (Zhuge et al., 2024) | Spikformer-4-384 | - | 4 | - | 95.2/77.9 |
| STAS$^\epsilon$ | Spikformer-4-384 | 9.32 | 4 | **0.40/0.50** | **95.2/77.9** |
| OST (Song et al., 2024) | OST-4-384 | 11.37 | 1 | 0.46 | 95.6/78.8 |
| Spikingformer (Zhou et al., 2023) | Spikingformer-4-384 | 9.32 | 4 | **0.42*/0.50*** | **95.7*/79.4*** |
| STATA (Zhuge et al., 2024) | Spikingformer-4-384 | - | 4 | - | 95.8/79.9 |
| STAS$^\epsilon$ | Spikingformer-4-384 | 9.32 | 4 | **0.37/0.46** | **95.8/79.4** |

for SNN-based ViTs, we also compare our results with those of OST and STATA. We measured the energy consumption[3] and accuracy of each model during inference on ImageNet (in Table 2) and CIFAR-10/CIFAR-100 (in Table 3).

**ImageNet** We trained STAS on the Spikingformer-8-384 and Spikingformer-8-512 models. We set hyper-parameters as $\alpha = 5$, $\beta = -25$, and $\delta_p = 10^{-4}$. To compare against a static token-dropping method, we implemented STATA[4] and evaluated its performance. As shown in Table 2, Transformer-based methods generally outperform CNN-based ones. On the Spikingformer-8-384, STATA reduces some energy but incurs a significant accuracy drop because it drops a fixed ratio of tokens without considering timesteps. In contrast, STAS reduces energy consumption while achieving even higher accuracy than the original Spikingformer. Furthermore, by adjusting the halting threshold $\epsilon$, we can create a variant, STAS$^\epsilon$, which trades some accuracy for greater energy savings. When configured

---

[3]Following the widely accepted measurement methods in previous SNN studies (Zhou et al., 2022; 2023), the equation for calculating energy consumption is provided in the supplement.

[4]Same as Footnote 1.

Table 4: Ablation study on Spikformer-4-384 and Spikingformer-4-384 with CIFAR-100.

| Architecture | I-SPS | $\epsilon$ | Accumulation | Avg. tokens | Acc (%) |
|---|---|---|---|---|---|
| Spikformer | ✗ | ✗ | Ⓑ | ×0.60 | 78.0 |
| | ✗ | ✗ | Ⓣ + Ⓑ | ×0.63 | 77.3 |
| | ✓ | ✗ | Ⓣ + Ⓑ | **×0.46** | **78.1** |
| | ✓ | ✓ | Ⓣ + Ⓑ | **×0.42** | **77.9** |
| Spikingformer | ✗ | ✗ | Ⓑ | ×0.65 | 78.5 |
| | ✗ | ✗ | Ⓣ + Ⓑ | ×0.95 | 77.4 |
| | ✓ | ✗ | Ⓣ + Ⓑ | **×0.70** | **79.9** |
| | ✓ | ✓ | Ⓣ + Ⓑ | **×0.50** | **78.5** |

for significant energy savings, STAS reduces the energy consumption of the original Spikingformer by 18.8% to 30.1% while maintaining a comparable or even slightly higher accuracy.

**CIFAR-10/CIFAR-100**   We trained STAS on Spikformer-4-384 and Spikingformer-4-384. We set hyper-parameters as $\alpha = -5, \beta = 0, \delta_p = 10^{-3}$ for Spikformer, and $\alpha = 5, \beta = -25, \delta_p = 10^{-3}$ for Spikingformer. For a fair comparison, we adjusted the halting threshold $\epsilon$ to create STAS variants tuned to the accuracy levels of the original models. For the Spikformer, we achieved substantial energy reductions of 45.9% on CIFAR-10 and 43.8% on CIFAR-100, respectively, while attaining higher accuracy. On the Spikingformer, STAS also achieved higher accuracy while reducing energy by 11.9% on CIFAR-10 and 8.0% on CIFAR-100.

### 4.3   ABLATION STUDIES

We evaluate the impact of I-SPS and the accumulation methods on the accuracy and energy efficiency of STAS. Table 4 shows the average number of tokens used per block and the corresponding accuracy with and without each component. All experiments are conducted on the Spikformer-4-384 model using the CIFAR-100.

**I-SPS vs SPS.**   Table 4 presents the token usage and accuracy of STAS with and without I-SPS. With I-SPS, STAS achieves higher accuracy (77.3% vs. 78.1%) while using fewer tokens (×0.63 vs. ×0.46). This improvement arises because, as shown in Fig. 3(c), I-SPS encourages similarity among inputs across consecutive timesteps, enabling more efficient application of ACT.

**Two- vs one-dimensional halting.**   Table 4 compares the halting score accumulation methods on CIFAR-100: one that accumulates only across one dimension (Ⓑ, block-level only) and another that accumulates scores across two dimensions (Ⓣ + Ⓑ, both timestep and block-levels as per Eq. equation 4). As shown in Table 4, the two-dimensional halting mechanism achieves higher accuracy (78.0% vs 78.1%) while removing more tokens (×0.60 vs ×0.46) compared to the one-dimensional halting. This is because, by definition, the LHS of Eq. equation 4 becomes larger under two-dimensional halting than under one-dimensional halting, which in turn increases the LHS of Eq. equation 9, leading to more tokens being halted. Furthermore, the STAS$^\epsilon$ variant maximizes this halting effect, achieving even greater token reduction (×0.42 and ×0.50 for Spikformer and Spikingformer, respectively).

## 5   CONCLUSION

In this paper, we addressed the fundamental two-dimensional (spatio-temporal) adaptive computation challenge inherent to SNN-based ViTs. We first identified that the efficacy of dynamic halting is fundamentally constrained by the static architecture's lack of temporal similarity. To resolve this, we proposed STAS, a framework that co-designs a static architectural module (I-SPS) with a dynamic halting policy (A-SSA) to enable accuracy-aware token halting across both spatial and temporal axes. Our experiments on CIFAR-10, CIFAR-100, and ImageNet demonstrate the effectiveness of this synergistic approach: STAS significantly improves the accuracy-energy trade-off, reducing energy consumption by up to 45.9%, 43.8%, and 30.1%, respectively, while simultaneously enhancing accuracy.

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

APPENDIX

This document provides supplementary material to the main submission. Sec. A details a widely accepted equation (Chen et al., 2023) for calculating SNN energy consumption and discusses the minor runtime overhead of our halting mechanism. We then present a detailed analysis in Sec. B on the impact of key hyperparameters ($\delta_p$, $\alpha$, and $\beta$) and surrogate module learning. Sec. C evaluates the generalizability of STAS on dynamic vision sensor (DVS) datasets (Li et al., 2017), and Sec. D provides additional qualitative results visualizing the token halting process.

## A ENERGY CALCULATION

To measure the energy consumption of an SNN, we calculate the theoretical energy usage based on the number of operations during inference. To do this, we first define the number of synaptic operations in each block as follows:

$$\text{SOPs}(l) = T \times fr(l) \times \text{FLOPs}(l), \tag{11}$$

where $l$ represents the index of the block, and $T$ denotes the timestep. The term $fr(l)$ refers to the firing ratio of spikes entering block $l$. $\text{SOPs}(l)$ indicates the number of synaptic operations performed in the $l$-th block, while $\text{FLOPs}(l)$ denotes the number of floating-point operations in the same block. Using SOPs, we can calculate the total energy consumption $E$ of the SNN as follows:

$$E = E_{MAC} \times \text{FLOPs}_{SPS} + E_{AC} \times (\text{SOPs}_{SPS} + \sum_{l=1}^{L} \text{SOPs}_{SSA}(l) + \sum_{l=1}^{L} \text{SOPs}_{MLP}(l)), \tag{12}$$

where $E_{MAC}$ and $E_{AC}$ represent the energy consumed per operation for multiplication and accumulation (MAC) and accumulation (AC), respectively, with $E_{MAC} = 4.6\text{pJ}$ and $E_{AC} = 0.9\text{pJ}$. $\text{SOPs}_{SPS}$ refers to the synaptic operations in the SPS, while $\text{SOPs}_{SSA}(l)$ and $\text{SOPs}_{MLP}(l)$ denote the synaptic operations in the SSA and MLP of a block, respectively. Additionally, $\text{FLOPs}_{SPS}$ represents the floating-point operations in the SPS. By preventing merged or masked tokens from firing spikes, STAS reduces the firing ratio $fr(l)$, reducing energy consumption in the SSA and MLP.

**Energy consumption for runtime overhead.** STAS performs additional computations at runtime to calculate the halting score for each token, which results in additional energy consumption. Since the computation for halting scores involves MAC operations, we estimate the energy per operation using $E_{MAC}$. Although halting scores are computed once per block and timestep, the operations are element-wise and lightweight, contributing only a negligible amount of energy compared to the total consumption of the model. For instance, STAS consumes at most only 0.03 mJ and 0.04 mJ of additional energy on ImageNet with Spikingformer-8-384 and Spikingformer-8-512, respectively, and just 0.005 mJ for each model on CIFAR-100 with Spikingformer-4-384. Note that the energy consumption of STAS reported in the main submission already includes all runtime overheads.

## B HYPERPARAMETER ANALYSIS

**Various $\delta_p$.** STAS allows for adjusting the trade-off between accuracy and the number of tokens through the hyperparameter $\delta_p$ in Eq. (10) in the main body of the paper. To examine the effect of $\delta_p$, we compare the accuracy and the number of tokens on the CIFAR-10 and CIFAR-100 datasets across a range of $\delta_p$ values from $10^{-1}$ to $10^{-4}$. We trained Spikingformer-4-384 during 410 epochs. Fig. 6 shows the accuracy and token usage during the training phase. As shown in Fig. 6, a smaller $\delta_p$ ($10^{-4}$, $10^{-3}$) results in higher accuracy, while a larger $\delta_p$ ($10^{-2}$, $10^{-1}$) leads to reduce the number of tokens. Consequently, STAS can be finely tuned by adjusting $\delta_p$ to achieve the desired balance between higher accuracy and fewer tokens, depending on the specific application requirements.

**Various $\alpha$ and $\beta$.** During training, STAS can control the trade-off between token usage and accuracy not only through $\delta_p$, but also via the hyperparameters $\alpha$ and $\beta$ in Eq.(3). The hyperparameters $\alpha$ and $\beta$ directly adjust the halting score, thereby influencing halting behavior during training.

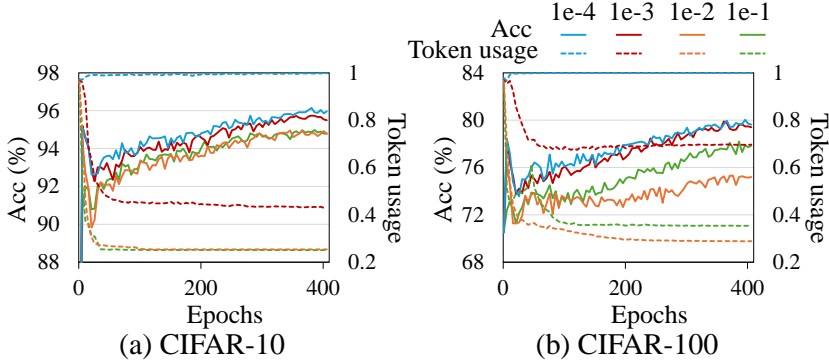

Figure 6: Training curve depedending on $\delta_p$ with Spikingformer

Table 5: Effect of $\alpha$ and $\beta$ on Spikingformer.

| Dataset | CIFAR-100 | | | | | | |
|---|---|---|---|---|---|---|---|
| $\alpha$ | 3 | 5 | 8 | $\beta$ | -15 | -25 | -35 |
| Avg. tokens | $\times 1$ | $\times 0.70$ | $\times 0.50$ | Avg. tokens | $\times 0.46$ | $\times 0.70$ | $\times 0.75$ |
| Acc (%) | 78.3 | 79.9 | 78.6 | Acc (%) | 78.7 | 79.9 | 79.9 |

Table 6: Effect of SML on Spikingformer.

| Dataset | CIFAR-10 | | | | CIFAR-100 | | | |
|---|---|---|---|---|---|---|---|---|
| SML | ✗ | | | ✓ | ✗ | | | ✓ |
| $\delta_p$ | 1e-3 | 1e-2 | | 1e-3 | 1e-3 | 1e-2 | | 1e-3 |
| $\epsilon$ | ✗ | ✗ | ✓ | ✗ | ✗ | ✗ | ✓ | ✗ |
| Avg. tokens | $\times 1$ | $\times 0.47$ | $\times 0.45$ | $\times 0.44$ | $\times 1$ | $\times 0.76$ | $\times 0.73$ | $\times 0.70$ |
| Acc (%) | 96.1 | 95.9 | 95.8 | 95.8 | 80.0 | 80.1 | 79.9 | 79.9 |

To investigate their effects, we conduct experiments on CIFAR-100, varying $\alpha \in \{3, 5, 8\}$ and $\beta \in \{-15, -25, -35\}$ while fixing $\delta_p = 10^{-3}$. We use Spikingformer-4-384, and all models are trained for 410 epochs. Table 5 shows the accuracy and average token usage across different $\alpha$ and $\beta$. As shown in the Table 5, increasing $\alpha$ results in lower token usage (e.g., $\times 1.00$ vs. $\times 0.50$). Conversely, decreasing $\beta$ also reduces token usage (e.g., $\times 0.75$ vs. $\times 0.46$), as it causes the halting scores to accumulate more rapidly.

**Surrogate module learning.** Surrogate module learning (SML) (Deng et al., 2023) effectively mitigates gradient errors during SNN training, thereby improving accuracy. Table 6 presents the effect of SML on token usage and accuracy under the setting of $\alpha = 5$ and $\beta = -25$ for both CIFAR-10 and CIFAR-100. As shown in Table 6, SML achieves reduced token usage (e.g., $\times 1.00$ vs. $\times 0.70$) while maintaining comparable accuracy (80.0% vs. 79.9%) on CIFAR-100 under the same setting. However, since the accuracy of STAS can be adjusted through hyperparameter tuning, we measure energy efficiency at comparable accuracy to SML by appropriately setting hyperparameters (e.g., $\delta_p$, $\epsilon$) for a fair comparison. Under these conditions, SML consistently demonstrates improved token efficiency at comparable accuracy. This suggests that training methods that enhance energy efficiency can be applied orthogonally to STAS without compromising accuracy.

## C  ADAPTABILITY OF STAS

**Another SNN-based transformer.** To verify whether our methodology works on ViTs based on directly trained SNNs other than Spikformer and Spikingformer, we applied it to spike-driven Transformer (Yao et al., 2023) and evaluated its performance on CIFAR-10 and CIFAR-100. We compared the accuracy of a model trained for 310 epochs with that of STAS, which was trained for an ad-

Table 7: Experiment results on Spike-Driven Transformer with four timesteps.

| Dataset | CIFAR-10 | | CIFAR-100 | |
|---|---|---|---|---|
| Method | STATIC | STAS | STATIC | STAS |
| Avg. tokens | ×1 | ×**0.38** | ×1 | ×**0.54** |
| Acc (%) | 95.6 | **95.8** | 78.4 | **78.9** |

Table 8: Experiment result on neuromorphic dataset with Spikingformer.

| Dataset | CIFAR10-DVS | | DVS128Gesture | |
|---|---|---|---|---|
| Method | STATIC | STAS | STATIC | STAS |
| Avg. tokens | ×1 | ×**0.60** | ×1 | ×**0.70** |
| Acc (%) | 81.3 | **82.4** | 98.3 | **97.9** |

ditional 310 epochs using the pretrained model. As shown in Table 7, similar to the results with Spikformer and Spikingformer, our approach maintains accuracy comparable to STATIC (without any lightweight method) in spike-driven Transformer, while reducing the average number of tokens used per block to 0.38 for CIFAR-10 and 0.54 for CIFAR-100.

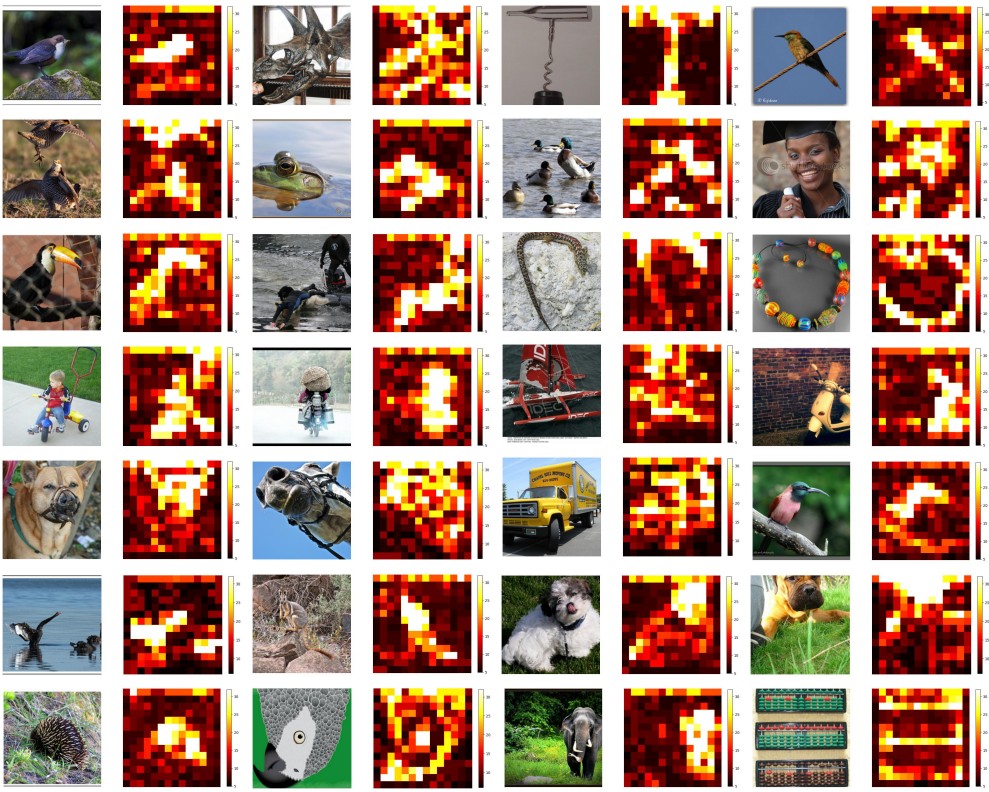

Figure 7: Original images (odd-numbered columns) and heatmaps showing the number of blocks (for four timesteps) each token processes (even-numbered columns) on ImageNet. Brighter colors indicate more processing per token. STAS halts earlier on tokens that lack visual information.

**Application to DVS Datasets.** To evaluate the adaptability of STAS, we tested its performance on the CIFAR10-DVS (Li et al., 2017) and DVS128Gesture (Amir et al., 2017) datasets. For these experiments, we trained a Spikingformer-2-384 model for 106 epochs with 16 timesteps, setting hyperparameters to $\alpha = 5$, $\beta = -10$, and $\delta_p = 10^{-3}$. As shown in Table 8, this configuration

still demonstrated strong performance, improving accuracy on CIFAR10-DVS to 82.4% with ×0.60 token usage, and maintaining comparable accuracy (97.9%) on DVS128Gesture with ×0.70 token usage. This highlights that while the full STAS co-design is optimal for static images, the A-SSA halting mechanism is robust and highly effective as a standalone module for processing inherently temporal data.

## D VISUALIZATION

We visualize STAS's token halting process on ImageNet samples using the Spikingformer-8-384 model. Figure 7 shows the original images alongside heatmaps that represent the computational depth of each token, defined as the total number of blocks it is processed for across four timesteps. Brighter colors in the heatmaps indicate more processing (later halting). The visualizations consistently show that STAS allocates more computation to tokens corresponding to salient object features. Conversely, tokens from uninformative regions, such as the background, are halted much earlier, resulting in darker areas on the heatmap. Notably, the policy appears more nuanced than simple foreground-background segmentation, often prioritizing semantically rich features within an object, like faces or distinctive textures.

## APPENDIX REFERENCES

Arnon Amir, Brian Taba, David Berg, Timothy Melano, Jeffrey McKinstry, Carmelo Di Nolfo, Tapan Nayak, Alexander Andreopoulos, Guillaume Garreau, Marcela Mendoza, et al. A low power, fully event-based gesture recognition system. In *Proceedings of the IEEE conference on computer vision and pattern recognition*, pp. 7243–7252, 2017.

Guangyao Chen, Peixi Peng, Guoqi Li, and Yonghong Tian. Training full spike neural networks via auxiliary accumulation pathway. *arXiv preprint arXiv:2301.11929*, 2023.

Shikuang Deng, Hao Lin, Yuhang Li, and Shi Gu. Surrogate module learning: Reduce the gradient error accumulation in training spiking neural networks. In *International Conference on Machine Learning*, pp. 7645–7657. PMLR, 2023.

Hongmin Li, Hanchao Liu, Xiangyang Ji, Guoqi Li, and Luping Shi. Cifar10-dvs: an event-stream dataset for object classification. *Frontiers in neuroscience*, 11:309, 2017.

Man Yao, JiaKui Hu, Zhaokun Zhou, Li Yuan, Yonghong Tian, Bo XU, and Guoqi Li. Spike-driven transformer. In *Thirty-seventh Conference on Neural Information Processing Systems*, 2023. URL https://openreview.net/forum?id=9FmolyOHi5.

