# OpenReview forum: "STAS: Spatio-Temporal Adaptive Computation Time for Spiking Transformers"
_ICLR.cc/2026/Conference — ICLR 2026 Conference Withdrawn Submission_

### Official Review · Reviewer_PxvN · 2025-10-23

**Soundness:** 2
**Presentation:** 2
**Contribution:** 2
**Rating:** 2
**Confidence:** 4

**Summary:**

The authors approach the problem from the perspective of dynamic computation, analyzing the current challenges in the field of SNNs—specifically, the difficulty of performing dynamic computation simultaneously in both spatial and temporal dimensions. They identify the root causes as architectural constraints and low temporal similarity, and ultimately achieve score-based early halting across both space and time through coordinated architecture and module design, thereby realizing truly spatio-temporal dynamic computation.

**Strengths:**

1. The analysis of dynamic computation–related content is relatively comprehensive.

2. Most of the research methods are explained clearly and directly through mathematical formulations.

3. The experimental analysis is also sufficiently detailed, occupying the majority of the paper’s content.

**Weaknesses:**

1. The writing order and the layout of figures and text in the paper are confusing. The Introduction repeatedly mentions the principle of Adaptive Computation Time (ACT) and seems to treat it as the theoretical foundation of the paper, yet the first actual explanation of what ACT is does not appear until the Related Works section. This leaves readers puzzled while reading the introduction. The layout and distribution of Figure 1(a) are also highly disorganized — the model distributions are overly dense, the labels unclear, and the figure is filled with numerous solid and dashed lines whose meanings are ambiguous, making it difficult to understand what the figure is intended to express. In Section 3, the authors propose two methods, I-SPS and A-SSA, but by the end of Section 3.1, they immediately present the ablation experiments and analysis for both modules, resulting in a very poor reading experience.

2. The paper suffers from insufficient citations and inadequate literature review and experimental comparison. In the first sentence of the Related Works section, the authors mention DT-SNN in the context of SNNs, but no citation is provided. The entire Related Works section is extremely short, consisting of only two paragraphs introducing dynamic computation in SNNs and ANNs, respectively. However, dynamic computation itself is a widely studied field. In the ANN domain, for instance, there is a 2021 survey paper[1] (currently cited over 1,000 times) that comprehensively analyzes the concept of dynamic neural networks, which the authors seem not to have reviewed. Moreover, the works discussed for ANN-based dynamic computation only extend up to 2022, revealing a clear lack of timeliness in the literature review — it is implausible that no further developments have occurred in the past three years.
Similarly, in the SNN domain, there exist multiple works addressing dynamic computation (including but not limited to [2,3,4]), yet the authors neither analyze the methods proposed therein nor clarify how they compare or differ from the present study.
Additionally, in Table 2, the authors include an entry labeled “ANN.” I assume this refers to the entire artificial neural network paradigm — yet there is only a single row of results, with no citations or explanations regarding the source of the data or its relevance. Furthermore, no performance comparison with other ANN-based dynamic methods is provided, which severely undermines the credibility of the results.

3. The motivation is insufficiently articulated, making the argument unconvincing. First, it is unclear why Transformer-based architectures are characterized as static—the authors never explicitly define what “static” and “dynamic” mean in this context. Isn’t the attention mechanism itself inherently dynamic? Second, the authors note that ACT was originally proposed for RNNs and state: “This concept was originally proposed for RNNs, where an encoder block B₁ iteratively refines its state from the same input x₁.” I fail to understand this reasoning: in RNNs, the input is an entire sequence, and each timestep corresponds to a different token, not the same input.
Furthermore, for SNNs, the low temporal similarity between timesteps could actually imply low redundancy in temporal information—each timestep dynamically attends to distinct aspects of the signal[4]. Forcing high temporal similarity might instead introduce temporal redundancy and lead to dynamic imbalance, contradicting the motivation for adaptivity.

4. Several technical details are unclear. For instance, I-SPS is only illustrated with a single figure, but no formal mathematical formulation is provided, making it difficult to understand how it actually works. Intuitively, it seems to eliminate the repetitive encoding used for static images, but the mechanism is not explicitly described.

5. Regarding ACT’s application in ViTs, my understanding is that it functions as a form of depth adaptivity, where layers are skipped based on computed halting scores. Why, then, is this interpreted as spatial adaptivity in the paper?

[1] Han Y, Huang G, Song S, et al. Dynamic neural networks: A survey[J]. IEEE transactions on pattern analysis and machine intelligence, 2021, 44(11): 7436-7456.

[2] Liu F, Zhao W, Chen Y, et al. Dynsnn: A dynamic approach to reduce redundancy in spiking neural networks[C]//ICASSP 2022-2022 IEEE International Conference on Acoustics, Speech and Signal Processing (ICASSP). IEEE, 2022: 2130-2134.

[3] Wang Q, Zhang T, Han M, et al. Complex dynamic neurons improved spiking transformer network for efficient automatic speech recognition[C]//Proceedings of the AAAI conference on artificial intelligence. 2023, 37(1): 102-109.

[4] Yao M, Richter O, Zhao G, et al. Spike-based dynamic computing with asynchronous sensing-computing neuromorphic chip[J]. Nature Communications, 2024, 15(1): 4464.

**Questions:**

See weaknesses.

---

### Official Review · Reviewer_JdfS · 2025-10-25

**Soundness:** 3
**Presentation:** 3
**Contribution:** 2
**Rating:** 6
**Confidence:** 5

**Summary:**

This study proposes STAS to address the unified application of ACT across both spatial and temporal dimensions in spiking Transformer. Specifically, STAS designs a static architectural module (I-SPS) and the Adaptive Spiking Self-Attention (A-SSA) to enhance the performance of the spiking Transformer. Extensive experiments on CIFAR-10/100 and ImageNet demonstrate that STAS reduces energy consumption significantly while improving accuracy compared to others.

**Strengths:**

1. The authors correctly identify the fundamental obstacle to applying ACT directly in spiking Transformers and provide visualizations and similarity analysis to support their claims.
2. The designs of the two components, i.e., I-SPS and A-SSA, are reasonable.
3. The paper conducts systematic experiments on the commonly used datasets of the SNN community and compares them against various SNN methods.
4. The article is clear and easy to understand.

**Weaknesses:**

1. I-SPS compresses spike inputs from multiple time steps into a single representation. While this design can improve temporal similarity and reduce the temporal cost, it discards information about dynamic changes across time steps. Are there any other compression methods that are not so aggressive? Authors should try such methods.
2. The parameter studies about different timesteps should be further explored, especially for single timesteps like OST and Att MS ResNet.
3. Missing the peer competitor in recent years, such as the Spike-driven Transformer series.
4. Does the proposed method require additional training costs compared to other direct training methods?

**Questions:**

The questions are provided in the Weaknesses part.

---

### Official Review · Reviewer_36if · 2025-10-30

**Soundness:** 2
**Presentation:** 2
**Contribution:** 2
**Rating:** 2
**Confidence:** 4

**Summary:**

This paper introduces STAS, a spatio-temporal adaptive computation framework for Spiking Transformers. The authors aim to bring Adaptive Computation Time---originally proposed for recurrent networks---into the domain of Spiking ViTs. To address temporal instability in SNN activations, the paper introduces I-SPS to enforce temporal smoothness, enabling A-SSA to dynamically halt computation along both spatial (token) and temporal dimensions. Experiments on CIFAR-10/100 and ImageNet demonstrate up to 45.9% energy reduction and improved accuracy.

**Strengths:**

1. The authors identify a valid gap—existing SNN-based Transformers perform redundant computation across time and tokens.
2. Combining early stopping (temporal) and token pruning (spatial) in one unified formulation.

**Weaknesses:**

1. The key insight---early exit or adaptive computation---is not new. It originates from Graves (2016, Adaptive Computation Time for Recurrent Neural Networks) and has since been explored in:
1) ACT and Adaptive Depth Transformers (Liu et al., DynamicViT, NeurIPS 2021; Rao et al., DynamicViT: Efficient Vision Transformers by Adaptive Token Sampling, NeurIPS 2021)
2) Token pruning and early exit in ViTs (Yu et al., FastFormer, NeurIPS 2021; Elbayad et al., Depth-Adaptive Transformer, ACL 2020)
3) Early stopping in SNNs (Stöckl & Maass, Optimized Spiking Neural Networks with Early Stopping, Front. Neurosci. 2021; Kim et al., Dynamic Temporal Sparsity for SNNs, ICLR 2022)
Therefore, the originality lies primarily in the adaptation of ACT to SNNs, not in inventing a new computational principle.
2. This paper does not provide a formal link between temporal smoothness (introduced by I-SPS) and the convergence property required by ACT. The mechanism remains empirical.
3. Overemphasis on energy efficiency without hardware validation.
4. Missing comparison to ANN-level dynamic ViT baselines.
5. The experiments are conducted entirely on frame-based datasets (CIFAR-10/100, ImageNet), while SNNs are typically evaluated on event-based datasets like DVS-CIFAR10, DVS128 Gesture, or N-Caltech101.

**Questions:**

See weakness

---

### Official Review · Reviewer_dtnF · 2025-11-01

**Soundness:** 3
**Presentation:** 3
**Contribution:** 3
**Rating:** 6
**Confidence:** 3

**Summary:**

The authors propose the STAS framework, which co-designs a static architecture with a dynamic computation strategy. Specifically, the I-SPS module integrates multi-timestep spike signals into a unified representation to establish temporal stability, while the A-SSA module leverages this stability to efficiently halt redundant token computations across both spatial and temporal dimensions. Experiments show that this method not only reduces energy consumption but also improves model accuracy.

**Strengths:**

1. This paper identify and solve a fundamental challenge in SNN-ViT models: since spike inputs vary across time steps, the model lacks temporal similarity, hindering the application of dynamic computation techniques such as ACT.

2. The design of the I-SPS module is both intuitive and effective, and the authors provide ablation studies that clearly validate its contribution.

3. The proposed method demonstrates impressive performance and efficiency across multiple datasets and model scales.

**Weaknesses:**

The A-SSA mechanism requires serial computation, accumulation, and checking of the halting score at each block and timestep. This serial checkpointing may introduce additional wall-clock latency, which should be examined through supplementary experiments.

**Questions:**

Why are the results for I-SPS only missing in Table 1, and the results for I-SPS and block-level only missing in Table 4? Including these results would provide a clearer understanding of the individual contributions of the I-SPS and A-SSA modules.

---

### Note · Authors · 2026-01-04

I have read and agree with the venue's withdrawal policy on behalf of myself and my co-authors.